# Improvement of the Solubilization and Extraction of Curcumin in an Edible Ternary Solvent Mixture

**DOI:** 10.3390/molecules26247702

**Published:** 2021-12-20

**Authors:** Verena Huber, Laurie Muller, Johnny Hioe, Pierre Degot, Didier Touraud, Werner Kunz

**Affiliations:** Institute of Physical and Theoretical Chemistry, University of Regensburg, D-93040 Regensburg, Germany; lauriemuller4@gmail.com (L.M.); johnny.hioe@ur.de (J.H.); pierre-emile-lucien.degot@ur.de (P.D.); didier.touraud@chemie.uni-regensburg.de (D.T.)

**Keywords:** natural deep eutectic solvents, curcumin, green solvents, solubilization, extraction, surfactant-free microemulsions

## Abstract

A water-free, ternary solvent mixture consisting of a natural deep eutectic solvent (NADES), ethanol, and triacetin was investigated concerning its ability to dissolve and extract curcumin from *Curcuma longa* L. To this purpose, 11 NADES based on choline chloride, acetylcholine, and proline were screened using UV–vis measurements. A ternary phase diagram with a particularly promising NADES, based on choline chloride and levulinic acid was recorded and the solubility domains of the monophasic region were examined and correlated with the system’s structuring via light scattering experiments. At the optimum composition, close to the critical point, the solubility of curcumin could be enhanced by a factor of >1.5 with respect to acetone. In extraction experiments, conducted at the points of highest solubility and evaluated via HPLC, a total yield of ~84% curcuminoids per rhizome could be reached. Through multiple extraction cycles, reusing the extraction solvent, an enrichment of curcuminoids could be achieved while altering the solution. When counteracting the solvent change, even higher concentrated extracts can be obtained.

## 1. Introduction

*Curcuma longa* L. (*C. longa*) is a plant widely known due to the bright yellow pigments its rhizomes contain. These pigments are the so-called curcuminoids, polyphenols with a conjugated π-system. They can be found in contents spanning from 1–15% of curcuminoids per rhizome. Amongst the many representatives, the most prominent three, curcumin, which makes up approximately 71% of the total curcuminoids, demethoxycucumin (19%), and bisdemethoxycurcumin (9%), [1,2,3] were examined in this study. For reasons of simplicity, the words curcumin and curcuminoids are frequently used synonymously throughout literature. Hence, it will occur as well in this article.

Due to the numerous beneficial characteristics of the curcuminoids, such as their anti-cancer, antioxidant, anti-inflammatory, and antiviral [4,5,6] properties deeming the plant a superfood, the extraction of these active compounds from the plant material is extensively researched (approximately 400 publications in the past 10 years, according to Web of Science [7]). A lot of research, the results of which are already employed in industry, however, is based on non-sustainable or petroleum-based solvents such as methanol or acetone. Indeed, acetone is one of the best solvents known to humankind but lacks in terms of the 12 principles of green chemistry postulated by P. Anastas [8] and the 6 principles of green extraction after F. Chemat [9]. It is highly volatile and non-edible, i.e., it has to be removed after the extraction, resulting in waste. Therefore, alternative solvents, sometimes in combination with microwaves and ultrasound, have been studied. Among these alternative solvents, natural deep eutectic solvents (NADES) [10,11,12] and surfactant-free microemulsions (SFMEs) [10,13,14] can be found.

Surfactant-free microemulsions are solvent systems consisting of two immiscible phases, one hydrophilic (in most cases water) and one hydrophobic (usually an oil), which are mixed through the addition of a cosolvent. This cosolvent is usually a small amphiphilic molecule, such as in the case of ethanol. It is known that these kinds of ternary systems are powerful solubilization media. This ability of solving is due to mesoscale inhomogeneities or compartmentation, which can be imagined as fissures or aggregates, in the otherwise transparent solution. The so-formed oil or water domains behave similarly to classical microemulsions containing surfactants and can be observed via light scattering experiments. In these systems, hydrophobic solutes can be solved in the hydrophobic part of the SFME. Regarding the work of P. Degot et al. [13], the addition of water decreased the solubility of curcumin. Therefore, natural deep eutectic solvents were used as a hydrophilic alternative to avoid the loss of solubility.

Natural deep eutectic solvents are mixtures of naturally occurring, often small, and hydrophilic components such as amino acids, organic acids, or choline. Through strong intermolecular interactions, mostly hydrogen bonds, the resulting mixture has a strongly decreased melting point, sometimes below room temperature, with respect to their single components. Since the starting materials offer a great variety of possible NADES, they can be tailored to specific applications concerning their properties, such as solving ability or edibility. One of their drawbacks, however, is their high viscosity, which can be counteracted by the addition of liquids of lower viscosity or their use at elevated temperatures [11,15]. Especially for some extraction experiments concerning thermolabile compounds, higher temperatures are not an option, whereas incorporation into other solvents is.

To this purpose, in the previous study [10], a NADES based on choline chloride (ChCl) and lactic acid (Lac) was used as a replacement for water in an SFME to show the potential usefulness of NADES as part of extraction systems. Indeed, through incorporation into a ternary system, the viscosity issue could be brought under control and good extraction yields could be achieved.

In the present contribution, we extend our preceding study to a broad series of potentially promising NADES in terms of their ability to solve curcumin in ternary mixtures with ethanol (EtOH) and triacetin (TriA). The best NADES based on choline chloride was examined further regarding extraction experiments.

## 2. Results and Discussion

### 2.1. Examination of the Solubility Depending on Various NADES

In the preceding study, it was found that the addition of choline chloride + lactic acid (1:1 n/n) NADES to EtOH and TriA and their binary mixtures’ components leads to a significant increase in the solubility of curcumin [10]. In light of these results, ten other NADES, in addition to the ChCl+Lac NADES, were investigated based on their ability to enhance the solubility in the ternary systems. The chosen NADES are already known in literature [16,17,18,19,20] and were based on acetylcholine (AcCh), choline chloride (ChCl), and proline (Pro), which are compounds produced naturally in the body, combined with citric acid (Cit), lactic acid (Lac), levulinic acid (Lev), and malic acid (Mal). The structures of the used compounds can be viewed in Appendix B, Table A1. In Figure 1, the results of the UV–vis examination of the saturated 50:20:30 NADES/EtOH/TriA mixtures are shown. The ternary mixture of water/EtOH/TriA 50:20:30 (black line), the binary mixture of EtOH/TriA (40:60 *w*/*w*) (red line), and acetone (blue line) served as a reference. It is visible that all combinations containing NADES were by far (at least factor 6) superior to the ternary mixtures with water concerning the solubility of synthetic curcumin. This came as no surprise as it is widely known that curcumin is not soluble in water [1]. According to reports in the literature, NADES made of hydrophilic components, as opposed to water, are indeed able to solubilize hydrophobic phytochemicals, especially phenols via hydrogen bonding [21]. This already indicated that the solubilization of curcumin in surfactant-free microemulsions is limited by the aqueous phase. Thus, if a high solubility is desired, water should be replaced by natural eutectics.

In the binary EtOH/TriA mixture, a solubility synergy was found [13] at a composition of 40:60 (*w*/*w*), resulting in a 3-fold increase in curcumin solubility compared to the single components. Subsequently, comparing the ternary NADES mixtures with this binary EtOH/TriA mixture, it was found that only five of the tested NADES, one based on Pro and two based on ChCl and AcCh, could compete. Concluding from this observation, the power of the hydrogen bond acceptors could be ordered as AcCh > ChCl > Pro. Considering the structures of these molecules (cf. Appendix B, Table A1), it could be speculated that the charged quaternary nitrogen of AcCh and ChCl, as opposed to the secondary amine of Pro, could influence the higher solubility due to advantageous cation-π interactions. Already in 1996, interactions between cations and various aromatic systems had been reported. Weak non-covalent interactions such as these cation-π interactions must not be neglected. Most literature, so far, deals with interactions between either inorganic ions or alkylated ammonium ions with the aromatic side chains of tryptophan, phenylalanine, and tyrosine in proteins. They are strong enough for favorable molecular recognition as the ions interact with the quadrupole of the conjugated π-systems of aromatic rings [22,23,24,25]. Regarding the results of the NADES screening, the interactions of cations with aromatic amino acids seem to be comparable to those of the quaternary ammonium ions Ch and AcCh with the conjugated π-system of curcumin. Thus, through these interactions, a higher solubility of curcumin in NADES mixtures could be reached as compared to mixtures with Pro, which only possesses a non-charged, secondary amine.

The hydrogen bond donors in these five mixtures were Lac and Lev. The ChCl+Lac NADES, which has been investigated in the previous paper, showed a solubility that was almost twice as high as the solubility in the original binary mixture of EtOH/TriA (40:60) [10].

Even though this was a significant improvement, ChCl+Lac was not the best among the examined NADES as it could not surpass the solubility of curcumin in acetone, which is one of the best solvents known. Three NADES in the ternary mixtures were powerful enough to achieve a higher curcumin solubility compared to the solubility in acetone (>1.4 times more). These top three ternary NADES-solvent mixtures with ethanol and triacetin were AcCh+Lev > ChCl+Lev > AcCh+Lac, closely followed by acetone. Hence, it was concluded that levulinic acid, as a hydrogen bond donor, had the best positive effect on the solubility, followed by lactic acid. Additionally, levulinic acid is cheap and easily available. Since there are no harsh regulations concerning the use of Lev, a quantitative removal after the extraction is not necessary. NADES based on Cit and Mal were not powerful enough to create an environment that was beneficial for the solubility of curcumin. In conclusion, the power of the hydrogen bond donors could be ranked as Lev > Lac >> Mal ≈ Cit.

With regard to the standard error, the best NADES mixture with AcCh+Lev and the runner-up with ChCl+Lev show a similar solubility. Therefore, for further studies concerning the extraction of curcumin from the ground roots of *C. longa* for applications in the food sector, it was decided to examine the ChCl+Lev further. This NADES in the mixture with EtOH/TriA showed a high solving capacity, while ChCl is permitted in food contrary to AcCh, which is a neurotransmitter and is not approved to be used as a food additive. Therefore, after the extraction with the ChCl system, the removal of the extraction solvent is not necessary.

### 2.2. Phase Diagrams and Curcumin Solubility

To obtain a picture of the monophasic domain in the ternary mixture of ChCl+Lev/EtOH/TriA, a phase diagram was recorded, which is shown in Figure 2a. To be accurate, the regarded system was rather a pseudo-ternary system than a ternary one, as it consisted of four components in total, since the NADES was a two-component system, thermodynamically speaking. For reasons of simplicity, however, the NADES was regarded as a unit and the system will also be referred to as a ternary mixture. As in the two already reported phase diagrams of ChCl+Lac/EtOH/TriA and water/EtOH/TriA (Figure 2b,c), one large two-phase region was determined between the hydrophilic part (ChCl+Lev) and the hydrophobic part (TriA). EtOH served as the cosolvent for all three ternary mixtures. One very noticeable difference in the newly established phase diagram is that ethanol is not completely miscible with the hydrophilic phase of ChCl+Lev. No more than 65 wt% of the NADES could be solved in ethanol. This also explains why the biphasic domain in the ChCl+Lev system is by far larger than in the preceding systems. Approximately 35 wt% of EtOH are necessary to ensure complete miscibility of the three components, as compared to 25 and 20 wt% for the ChCl+Lac and water systems, respectively. This means that EtOH is not powerful enough to accomplish a better miscibility of the levulinic acid-based NADES with TriA. It can be assumed that a cosolvent that is able to fully solve the NADES, while still being miscible with TriA, could decrease the two-phasic area. It is possible to think of mixtures of water and EtOH, glycerol, or PEG. However, it has to be kept in mind that changing the cosolvent will have an influence on the preferential solubility of curcumin in the binary of cosolvent and TriA, which has been described by P. Degot et al. [13]. The miscibility of the three compounds ChCl+Lev, EtOH, and TriA, however, was enough to form a monophasic domain of interest concerning extraction experiments without overcomplicating the system.

To check the existence of aggregations that would justify the classifications of these systems as SFMEs, dynamic light scattering experiments were performed. Through these experiments, the reminiscent critical points (RCPs) of the underlying binary mixtures of NADES/EtOH and EtOH/TriA and the critical points (CPs) of the ternary mixtures were determined. CPs are the points where the critical fluctuations are highest; thus, close to the points, the correlation functions, as an indication of structuring, should reach their maximum [26,27]. The term reminiscent critical point, in this case, was introduced to describe the critical points of the binary that are observable at room temperature. Since the binary solutions involving EtOH are mostly monophasic, one cannot really talk about critical points as this would imply a phase separation. However, light scattering signals, indicating fluctuations and solving anomalies in these fully miscible solutions, suggest that at certain conditions (in the present case temperature) the binary solutions will demix. The compositions where the demixing will happen first would mark the critical point. The projection of this composition to room temperature where fluctuations were visible were then regarded as the RCPs.

Several samples close above the two-phasic region were examined and the corresponding correlograms can be seen in Appendix A of the Appendix A. In view of the curves of Appendix A, the critical points of the ternary systems could be established on the hydrophilic side. Considering Appendix A, the correlation curves obtained in the ChCl+Lev system are rather indistinctive. Only a slight rise is visible at around 20 wt% of TriA. The correlation curves in the system with ChCl+Lac were more pronounced and a critical point at around 40 wt% of TriA was found. Only in the classical surfactant-free microemulsion consisting of water/EtOH/TriA, well-defined correlation functions were found, and the critical point was determined to be situated at a TriA content between 30 and 40 wt%. The obtained critical points (CPs) are marked in turquoise in the ternary phase diagrams. Since the correlation functions are only slightly pronounced, indicating no detectable aggregates such as micelles but only slight fluctuations, it could be assumed that the NADES systems exhibit a classical critical point. Further, it seems that the ChCl+Lev/EtOH/TriA system has a very good solubilization power without being an SFME, as it does not exhibit significant aggregate forming.

Additionally, the binary mixtures NADES/EtOH and EtOH/TriA were investigated. The respective correlation functions can be seen in Appendix A of the Appendix A. The mixture of water and EtOH has not been measured as it has already been studied by T. Buchecker and S. Krickl et al. [28] in 2017. They did not find significant correlations depending on the EtOH content of the binary mixture, even though it is known that there are mixing inhomogeneities in these media. Due to their high diffusion coefficients, they produce too fast fluctuations which cannot be recorded using DLS. Therefore, no RCP for the mixture of water and EtOH was determined. In the binary mixtures containing the NADES, ChCl+Lev/EtOH and ChCl+Lac/EtOH (Appendix A respectively), an increase in correlation could be observed with an increasing NADES content. A maximum correlation between 40 and 50 wt% of ChCl+Lev was found shortly before phase separation. A similar trend was found with the ChCl+Lac NADES. However, at a NADES content of 50 wt%, the correlation functions turned bimodal. This behavior change was chosen to set the RCP. Thus, for both NADES systems, the RCP of the binary mixture was found at ~45 wt% of EtOH. For the binary mixture of EtOH/TriA, the correlations were only slightly pronounced, but the maximum correlation was found to be situated at 50 wt% of EtOH.

Of course, using DLS is no perfect way of determining the critical points, especially because the correlation functions of the NADES systems are only weakly pronounced. Regardless, a phase diagram of EtOH/TriA was simulated using COSMO-RS to support the findings of the light scattering experiments. To this purpose, EtOH/TriA mixtures were screened to find a critical temperature, below which the binary system was not completely miscible anymore. As EtOH and TriA are miscible at room temperature and above, it was assumed that an upper critical solution temperature (UCST), as defined by IUPAC, should be found [29]. It was hypothesized that the composition, where a phase transition occurred at the critical temperature, and the composition where the highest detectable correlation functions at a different temperature were found, should coincide. So, at this composition of the critical point of the binary system, the “reminiscent critical point” should also be situated. From this critical point at low temperatures, the reminiscent critical point at room temperature could be inferred.

However, performing the simulation was also difficult for this system, as severe convergence issues were encountered during the calculation of the liquid–liquid equilibria (LLEs). Therefore, only crude estimations of these LLEs were currently possible. The explanation would exceed the scope of this article. Therefore, the explanation along with the preliminary phase diagram can be viewed in the Appendix A (cf. Appendix A).

Nevertheless, when reducing the temperature to −30.0 °C, two points of liquid–liquid equilibrium at EtOH/TriA ratios of 65:35 and 80:20 were found. This indicates that the simulation reached a biphasic system. In contrast, the simulation at −20.0 °C showed no LLE points at any mixture of EtOH/TriA. Hence, most likely, a UCST was encountered at relatively low temperatures ranging between −20.0 and −30.0 °C. This means, above a temperature of −30.0 °C, the liquids ethanol and triacetin are miscible at any proportion. Hence, the reminiscent critical point can be described as the shadow of the critical point at the UCST, which lies below −20.0 °C, that is still visible through slight fluctuations detectable via DLS at room temperature. Even though the results of DLS and the simulation differ, which is due to calculation difficulties and differences in temperature, of course, it was found that there is some kind of solving anomaly, such as fluctuations, in the binary of EtOH/TriA.

In general, critical fluctuations might act like nano-fissures in the solvent. These fissures can be starting points for changes in the liquid, such as phase separation, which was simulated between EtOH and TriA. Through the high speed of formation and deformation, favorable interactions between the solvent and newly introduced compounds are possible, leading to a preferential solubility of the latter, in this case, curcumin. As the solvent molecules only slightly interact with each other, curcumin can change the solvent organization drastically in contrast to the solubilization, which is observed in classical surfactant systems and was already described by P. Degot et al. [30]. Using UV–vis measurements and COSMO-RS calculations, P. Degot et al. [13] found a solubility synergy that is located close to the critical composition of EtOH and TriA in Figure 2c. The RCP was set at the point where the highest fluctuations were found via DLS. The RCP simply represents the composition, where phase separation between the two miscible liquids would occur at the critical temperature. UV–vis measurements conducted during this study also showed that, especially in the NADES systems, the maximum solubility of curcumin was close to the CPs and RCPs. This indicated that the reminiscent critical points, where the solubility was highest in the binary mixtures, had an influence on the position of the critical point in the ternary systems. This is analogous to what P. Baudin et al. [31] reported when adding short propylene glycol alkyl ethers to microemulsions. The characteristic of the propylene glycol alkyl ethers to change the demixing temperatures of the microemulsions was used to tune the formulation in the desired way. In that case, the temperature dependence is induced by the formulation, contrary to the present case, where temperature sensibility is inherent. The name reminiscent critical point was chosen to figuratively describe that the critical points of the binary systems were still in the “memory” of the liquid when creating the ternary mixtures.

Regarding only the solubility of curcumin in the ternary systems, with every change of the hydrophilic part, the curcumin solubility could be improved (ChCl+Lev > ChCl+Lac > H_2_O). This is in accordance with the results presented in Figure 1, showing that all NADES combinations improve the curcumin solubility. By using a choline-based NADES, the area of maximum solubility is shifted away from the binary EtOH/TriA mixture to the ternary, from the RCP towards the CP of the ternary mixture. When using ChCl+Lev as the hydrophilic phase of the ternary mixture, the maximum solubility of curcumin could be improved by a factor of ~2 and 7 with regard to the best compositions in the ChCl+Lac and water systems respectively. The acid of the NADES compounds, however, still has an influence on the manner of solubilization. The solubility improvement can be explained as follows:

By looking at the diagrams, three different ways of solubilization can be recognized. Considering the mixture of EtOH with TriA, the improvement can be explained through preferential solvation. The solubility compared to EtOH only was improved in the mixture by a factor of 6, which amounts to an energy gain of approximately 1.8 *k_B_T*. The maximum solubility in the water-based system is not in the ternary but in the binary, far away from the CP. Of course, as described above, the fluctuations of the solvent at the RCP composition might have an influence, but this preferential solubilization at room temperature means that, in this case, the solubility is entropically driven and not by structuring [32].

In the system with ChCl+Lac, the highest solubility is due to critical fluctuations of the whole ternary system and not due to preferential solvation. This is visible since the optimum solubility of the phase diagram in Figure 2b is located close to the critical point (CP). In comparison to pure EtOH, a solubility improvement of approximately a factor of 10, translating to a gain in energy of 2.2 *k_B_T*, was found. In combination with the DLS curves (cf. Appendix A), one can see that, for this kind of solubilization, a defined pre-structuring is hardly important but rather fluctuations of the global network [33].

When looking at Figure 2a, the ternary system with ChCl+Lev, it is visible that the acid of the NADES has an influence on the way curcumin is solved. As already said above, this system shows the highest solubility of curcumin with an energy gain of 3.5 *k_B_T*. The domain of highest solubility is also close to what was described as the CP in this system; however, the distance is bigger than in the ChCl+Lac system. Hence, one can assume that the solubilization in this system is not so much depending on critical fluctuations but rather on the formation of a surfactant-free microemulsion with so-called pre-Ouzo structuring [34,35]. To be sure of these assumptions, further experiments of neutron scattering have to be conducted. However, this would exceed the scope of this study as the focus was only on finding solvent compositions with superior solubilization power. In light of the solubility improvement in the ChCl+Lev system, extraction experiments were conducted.

### 2.3. Extraction

To examine the extraction ability, three different points in the phase diagram have been investigated. One point in the binary mixture of ChCl+Lev/EtOH (55:45), close to the reminiscent critical point, and two points in the ternary mixtures of ChCl+Lev/EtOH/TriA where the solubility was at a maximum (30:40:30) and along the dilution of the best binary EtOH/TriA composition with ChCl+Lev at (ChCl+Lev/EtOH/TriA 25:30:45), respectively. The latter was chosen specifically, as it was close to the two-phasic region and showed a high solubility.

According to the preceding work by V. Huber et al. [10], a powder to solvent ratio of (1:8) rhizomes to extraction solvent in weight was chosen. Since the mixtures are very similar, an exchange of Lac with Lev should not have a significant influence on the necessary ratio.

Samples were prepared in triplicates and the curcuminoid content was analyzed using HPLC. The quantification of the curcuminoids was conducted via external calibration of the three curcuminoids. The calibration curves are shown in Appendix A of the Appendix A.

The Soxhlet results, as obtained by P. Degot et al. [13], were used as a first reference. The process of the exhaustive extraction, however, was improved in this study by modifying a Soxhlet apparatus with an oil pump and a valve to have pressure control. Through a lowered pressure, the boiling point of acetone was reduced, and it was possible to extract the thermosensitive curcuminoids more gently at a temperature of only 30 °C. The results of all extraction experiments are given in Table 1. As can be seen, when using the softer method for the exhaustive extraction, a total of 5.3 mg curcuminoids per gram rhizomes could be extracted. This means that 20% more curcumin, 10% more demethoxycurcumin, and 56% more bisdemethoxycurcumin were extracted. This again confirms that bisdemethoxycurcumin is the most labile of the investigated curcuminoids, for without the pressure and temperature control more than half of its amount was degraded. Of course, even with the lower temperature, it is not completely certain that no thermo-degradation occurred and the total lot of curcuminoids was extracted. However, the great improvement at the lower temperature of 30 °C, as opposed to >56 °C (boiling point of acetone at atmospheric pressure), attests to the fact that, with the pressure controlled Soxhlet, a truer value of the total curcuminoid amount could be obtained. Hence, the yields of the pressure controlled Soxhlet extraction will be used as a 100% reference.

The ChCl+Lev/EtOH/TriA system yielded the highest extraction results (~84%) in comparison to all the references (except for the pressure controlled Soxhlet) of the best extractions of the previous studies [10,13]. This means that ChCl+Lev is a superior hydrophilic extraction component compared to ChCl+Lac (~79%) and water (~68%). This can be attributed to the improved solubility of curcumin in the Lev-based NADES.

In the binary mixture of NADES/EtOH (55:45), the curcumin solubility is roughly 25% lower than in the ternary optimum composition of NADES/EtOH/TriA (30:40:30). The extraction results, however, are almost identical (difference of 0.13 ± 0.07 mg per g rhizome). Apparently, the extraction power did not only come from the solubility of curcumin in the solvent but also from the interactions with the plant material, specifically, how the solvent penetrates the plant matrix and how the solutes, here the curcuminoids, are transported out of the ground rhizomes [36,37].

For further experiments of the cycle extraction, based on the previous research by P. Degot et al. [14], the optimum composition was used to exploit the full potential of its high solving capability. Since the solubility was almost twice as high in this composition compared to the extraction mixture based on the Lac NADES [10], it was estimated that 12 cycles of extraction should be possible. The results are presented in Figure 3. The concentrations of the enriched tinctures can be viewed in Appendix A of the Appendix A. One should not mistake the total amount of extracted curcuminoids (as depicted in Figure 3) for the concentration (depicted in Appendix A). In Figure 3, the total mass of curcuminoids can be viewed without regard to the loss of volume that was bound to happen, as a part of the solvent stayed behind in the plant material and could not be removed. The concentration is only an indication of the curcumin amount in the remaining solvent.

The extraction mixture could indeed be enriched up to seven cycles of extraction with a total curcuminoid content of ~180 mg in the solvent. This is the same number of extraction cycles and amount of curcuminoids that could be reached in the reference mixture containing the Lac NADES. In this figure, only the total amount of the extracted curcuminoids is presented. The concentrations are presented in Appendix A of the Appendix A. Looking at Figure 3b, the blue and red curves superpose each other, indicating that ChCl+Lev, even though yielding higher extraction results in a single extraction cycle, did not improve the cycle extraction.

To see if a premature saturation was reached, two samples containing the synthetic chemicals were prepared. One sample only contained curcumin to imitate the total curcuminoid amount after seven cycles (green down-facing triangle) and the other contained curcumin and bisdemethoxycurcumin (pink diamond). Twice the amount of bisdemethoxycurcumin that could be obtained through extraction was used to imitate the total content of demethoxy- and bisdemethoxycurcumin due to the high cost of demethoxycurcumin. Figure 3b shows that the composition of the curcuminoids hardly had any influence on the following extraction of the ground rhizomes. Indeed, it was possible to enrich the artificially prepared samples with more curcuminoids, as was expected according to the UV–vis results.

This, of course, raised the question of why the extraction solvent could not be enriched further in the classic way of reusing the solvent.

A small improvement up to 10 extraction cycles was still possible, surpassing the reference of the saturated cycle 7 of the ChCl+Lac system [10]. However, the intensity of the increase was smaller (cf. Figure 3a). This was an indication that a slow saturation happened. Additionally, such a small increase does not justify the experimental efforts of three more cycles. Therefore, cycle 7 can be considered as the final, saturated one. As mentioned before, approximately 12 extraction cycles could be estimated at such a high solubility. However, the curcuminoid content decreased when performing 12 cycles. This phenomenon also occurred for P. Degot et al. [14]. The saturation occurred after three cycles and the amount of curcuminoids decreased slightly after the 4th cycle of extraction. This is counterintuitive and will be investigated in the following:

Since one cycle of extraction takes approximately 1 h to be finished, the extraction of multiple cycles demands a lot of time (up to 12 h for all 12 cycles). Therefore, it is possible that the extraction solvent changed over time. Since TriA contains three ester bonds and the NADES is made of an acid, it was assumed that an acidic ester hydrolysis might have taken place. To check this, ^13^C-NMR samples were recorded over three days. The spectra can be viewed in Appendix A of the Appendix A. Assuming that hydrolysis had taken place, two new peaks at around 20 and 175 ppm representing acetic acid should have appeared. The NMR spectra, however, did not change at all over the three days. Hence, the possibility of acidic hydrolysis was eliminated. Moreover, the EtOH peaks changed neither in position nor intensity. Its evaporation could also not be the reason for the solvent change.

Another possibility to monitor changes in the solvent were conductivity measurements, as the choline chloride of the NADES is an electrolyte. To this purpose, a calibration curve of the conductivity in the solvent with a changing NADES content was recorded (see Figure 4a). The conductivity was monitored by diluting the optimum composition of ChCl+Lev/EtOH/TriA (30:40:30) with a mixture of EtOH/TriA (40:60). As depicted in Figure 4a, the conductivity decreased with a decreasing amount of NADES as fewer charges were present in the solution. Thus, a maximum conductivity of ~4.3 mS/cm was found at the original optimum composition.

Then, extraction experiments were performed, and the conductivity was recorded after each cycle. In Figure 4b, it is shown that after each cycle of extraction, the conductivity decreased significantly down to ~3.7 mS/cm after seven cycles of extraction. Comparing this to the results of the calibration curve, the NADES content was decreased to ~25 wt% in the mixture. A possible explanation for this change in the solvent conductivity and therefore composition is the adsorption of choline to the plant material. Of course, like all plants, *C. longa* also contains dietary fibers such as cellulose. According to literature, the negatively charged cellulose residues and the choline cation share strong ionic interactions [38,39]. Due to these interactions, the ChCl would stay in the remaining plant material after the extraction. For a new extraction cycle, the old rhizomes were removed from the solvent and replaced by fresh ones, altering the solvent composition. This means that the composition of the extraction mixture was changed to such an extent that the solubility of curcumin decreased significantly. Hence, a slow saturation starting at cycle 7 and continuing up to cycle 10 was observed. The loss of curcuminoids in cycle 12 can then be easily explained as well: the solubility decreased to such an extent that the previously extracted curcuminoids could not be solved in the solution anymore, and thus stay behind. Therefore, the amount of curcuminoids decreased.

When monitoring the conductivity of the solvent and compensating for the loss of NADES throughout all extraction cycles, it should be possible to exploit the whole potential of the solvent to perform the 12 cycles of extraction analogously to the imitated samples of Figure 3b.

## 3. Materials and Methods

### 3.1. Materials

The synthetic phytochemical curcumin (purity > 97%) and bisdemethoxycurcumin (purity > 98%) were bought from TCI (Eschborn, Germany), demethoxycurcumin (purity > 98%) from Sigma-Aldrich (Darmstadt, Germany). The dried rhizome powder of *C. longa* was purchased from Kwizda (Linz, Austria).

The hydrogen bond acceptors acetylcholine chloride (AcCh, purity > 99%), choline chloride (ChCl, purity > 99%) were bought from Sigma-Aldrich (Darmstadt, Germany) and proline (Pro, purity > 99%) was bought from TCI (Eschborn, Germany). The hydrogen bond donors citric acid (Cit, purity > 99.5%), lactic acid (Lac, purity > 85%, FCC), levulinic acid (Lev, purity > 98%) were obtained from Sigma-Aldrich (Darmstadt, Germany), malic acid (Mal, purity > 99%) from Alfa Aesar (Thermo Fisher GmbH, Kandel, Germany), and oxalic acid dihydrate (Ox, purity > 99%) from Merck (Darmstadt, Germany). To see the corresponding structures and abbreviations of the hydrogen bond donors and acceptors, regard Table A1 of the Appendix B.

The solvents ethanol (EtOH, purity > 99%), acetonitrile (purity > 99%, HPLC grade), acetic acid (purity > 99%), and triacetin (TriA, purity > 99%, food-grade) were purchased from Merck (Darmstadt, Germany). Water was deionized using a Millipore Milli-Q purification system (Merck Millipore, Billerica, MA, USA).

All chemicals were used without further purification.

### 3.2. Methods

#### 3.2.1. NADES Preparation

In this study, the wide choice of NADES was narrowed down by only regarding 1:1 molar mixtures for better comparability of the results.

A heating method was used to prepare the NADES. Therefore, the hydrogen bond acceptors and hydrogen bond donators, the organic acids, were mixed in a mortar at the respective molar ratio and heated to 80 °C until a clear liquid was obtained (~90 min) [10,40]. The mixtures were left to cool down to room temperature before use.

#### 3.2.2. Solubility Measurements

For the NADES screening, 5 g samples of the ternary NADES/EtOH/TriA were prepared with a weight distribution of 50/20/30, respectively. This composition was chosen as it is a dilution of the optimum binary mixture of EtOH/TriA (40:60). Additionally, the effect of NADES could be viewed more closely at a high share. The samples were saturated with synthetic curcumin and left to stir at room temperature for 1 h to ensure saturation.

The contour diagram, representing the solubility of curcumin in the monophasic domain, was recorded for the system ChCl+Lev, which was the NADES of choice, EtOH, and TriA. To qualitatively examine the solubility, 5 g of binary and ternary mixtures of NADES/EtOH, EtOH/TriA, and NADES/EtOH/TriA were prepared and saturated with synthetic curcumin.

All the samples were filtered through 0.45 µm PTFE filters to remove the excess curcumin. Successively, the solutions were examined via UV–vis spectroscopy, comparing the maximum absorbances at λ_max_ = 425 nm. The measurements were carried out using a Lambda 18 UV–vis spectrometer by PerkinElmer (Waltham, MA, USA). Solutions saturated with synthetic curcumin were examined to find the regions of highest solubility of curcumin in the monophasic regime of the ternary compositions. Before the measurement, all samples were diluted adequately with ethanol.

#### 3.2.3. Determination of the Ternary Phase Diagram

The ternary phase diagram, consisting of NADES/EtOH/TriA, was recorded by preparing 3 g of binary samples of NADES/EtOH and TriA/EtOH at different proportions in borosilicate tubes. TriA and NADES, as the respective third components, were added dropwise until the mixtures became turbid or showed precipitation. This phase separation was determined by the bare eye. To determine the miscibility gap, the added amount of the third component was recorded [10].

#### 3.2.4. LLE Simulation

For the simulation of binary phase diagram consisting of TriA/EtOH, COSMO-RS formulation within the software package COSMOTherm 2021. The LLE points were calculated with constant temperature and 0.05 increments of increasing EtOH concentration.

For triacetin, 3 experimental data points are available and recommended at 284.24, 298.27, and 318.20 K. In the case that at least 3 experimental data points are available, the Antoine equation with 3 fitting coefficients (Equation (1)) can be used. When at least 6 data points are known, which is true for ethanol, then the more sophisticated Wagner equation with 6 fitting coefficients (Equation (2)) can be applied.
(1)$$\ln\left(p_{i}^{0}\;\right)=A-\frac{B}{T-C}$$
(2)$$\ln\left(p_{i}^{0}\;\right)=\ln\left(A\right)-\frac{1}{1-\tau }\left(C\tau +D\tau^{1.5}+E\tau^{3}+F\tau^{6}\right)\;\mathrm{with}\;\tau =1-\frac{T}{B}$$

The total vapor mole fraction is calculated according to Equation (3)
(3)$$y_{i}=\frac{p_{i}^{0}x_{i}\gamma_{i}}{p_{tot}}$$

#### 3.2.5. Dynamic Light Scattering

To obtain an idea about the structuring of the solvents and to find the critical points in the binary and ternary, dynamic light scattering (DLS) experiments were performed using a temperature-controlled CGS-3 goniometer by ALV (Langen, Germany). The goniometer system was equipped with an ALV-7004/FAST Multiple Tau digital correlator and a vertically polarized 22 mW HeNe laser of a wavelength of λ = 632.8 nm. The solvent for analysis was filtered into dust-free cylindrical light-scattering cells of an outer diameter of 10 mm, using 0.2 µm PTFE syringe filters. The sealed samples were then thermostated in a toluene vat of 25 ± 0.1 °C. Each sample was measured for 120 s. The obtained correlation curves were evaluated only qualitatively. The same assumption that higher intercepts correspond to more pronounced and time-stable structuring, which was used by T. Buchecker and S. Krickl [28], was followed. Hence, the critical points were evaluated accordingly.

#### 3.2.6. HPLC Analysis

To analyze the curcumin content of the rhizome extracts of *C. longa,* a Waters HPLC system with two Waters 515 HPLC pumps, Waters 717 autosampler, and Waters 2487 UV–vis detector, and a Knauer Eurospher 5 C18-Column (100 Å, 250 × 4.6 mm) was used. The mobile phase was made up of water with 0.3% of acetic acid (A) and acetonitrile (B). All samples were eluted three times and also made in triplicates. The mobile phase started at an initial condition of 60% A and 40% B. After injection of the sample (10 μL), the mobile phase was first ramped up to 60% B over 17 min. Over 1 min, a mobile phase content of 100% B was reached, which was held for 6 min. Then, the mobile phase was ramped down to the starting condition of 40% B and 60% A over 1 min and held for 7 min. This gradient HPLC procedure, after P. Degot et al. [13], can be viewed in Appendix A of the Appendix A.

In a concentration range of 0.04 to 0.2 mg/mL, the calibration curves for the three curcuminoids were recorded, as shown in Appendix A in the Appendix A. The elution was carried out as described in this section.

#### 3.2.7. Pressure Controlled Soxhlet Extraction

The total curcuminoid amount in the rhizomes was determined via an exhaustive Soxhlet extraction using a classical Soxhlet apparatus enhanced with a pump and manually adjustable pressure control. To ensure no curcuminoids are lost through thermal degradation, the extraction was carried out at moderate conditions of 30 °C at a pressure of 234 mbar. A total of 2 g of ground rhizomes was extracted with 70–80 mL acetone. The extraction process was terminated after ~5 h as the extraction solvent surrounding the extraction thimble was visually transparent to the bare eye. This time corresponds to 55–60 cycles of extraction. Then, the extract was put in a volumetric flask and topped off with acetone, diluted 10-fold in ACN, and then eluted via HPLC. The extraction was carried out in triplicate.

#### 3.2.8. Extraction Procedure

A total 2 g of turmeric powder was extracted with 16 g of different extraction mixtures at room temperature under constant stirring at 1300 rpm for one hour. Then the extracts were centrifuged, the supernatant filtered, put in a volumetric flask, topped off with EtOH, diluted 25-fold in acetonitrile/water 90:10 (*w*/*w*) to prevent the precipitation of the NADES, and eluted by HPLC.

#### 3.2.9. Cycle Extraction

To enrich the solvent in curcuminoids, 2 g of rhizomes was extracted with 32 g of the extraction solvent at the optimum composition of 30:40:30 NADES/EtOH/TriA for one hour at 1300 rpm. The solutions were centrifuged at 4700 g for 10 min to separate the supernatant from the rhizome remnants (one cycle). For every further cycle, 2 g of fresh rhizomes was added to the collected supernatant and extracted again. After several cycles (up to 12), the supernatant was put into volumetric flasks, topped off with EtOH, diluted 25-fold in acetonitrile/water 90:10 (*w*/*w*), and eluted via HPLC. All samples were prepared in triplicate and the quantification was carried out by external calibration (cf. Appendix A of the Appendix A).

#### 3.2.10. ^13^C-NMR Analysis

An Avance III HD 400 NMR-spectrometer by Bruker (Billerica, MA, USA), which operates at 400 MHz, was used for the ^13^C-NMR measurements. Approximately 1 mL of the extraction solvent ChCl+Lev/EtOH/TriA (30:40:30) in weight was filled into Norell 507-HP-7 High-Precision-NMR sample tubes (Norell Inc., Morganton, NC, USA). The sample was measured right after preparation, one day, and three days after preparation. The sample tube was not sealed with parafilm, and the solvent was not changed to imitate the normal conditions the solvent would have to endure during cycle extraction.

#### 3.2.11. Conductivity Measurements

Conductivity measurements were conducted to determine if the extraction solvent changes over time and if a change is linked to the extraction. All measurements were performed in the temperature-controlled measuring cell (25 ± 0.2 °C) of a low-frequency WTW inoLab Cond 730 conductivity meter, connected with a WTW TetraCon 325 electrode (Weilheim, Germany) under permanent stirring (~500 rpm) and manual mixing, when necessary. Three different kinds of samples were examined.

First, 20 g of the optimum ternary composition (30:40:30 NADES/EtOH/TriA) was examined and diluted in 0.5–2 g steps with 100 g of a 40:60 (*w*/*w*) EtOH/TriA mixture.

Then, the acidic hydrolysis of TriA was imitated by replacing a part of the TriA with the respective amounts of glycerol and acetic acid. The conductivities of the artificially hydrolyzed samples were measured in the range of 0–100% of TriA hydrolysis.

Lastly, the conductivity of the extraction solvent after every cycle of extraction (up to 7 cycles) was determined.

## 4. Conclusions

In this study, different 1:1 (n/n) NADES based on choline chloride, acetylcholine, and proline were screened concerning their curcumin solving ability in ternary mixtures with ethanol and triacetin. It turned out that ammonium cations had a beneficial influence on the high curcumin solubility. The NADES ChCl+Lev, based on choline chloride and levulinic acid, was chosen for more detailed investigations.

A ternary phase diagram was recorded using the NADES, ethanol, and triacetin. Dynamic light scattering measurements were performed to find the critical points in the ternary system based on the ChCl+Lev NADES. Then, the critical point in the ternary could be strongly influenced by the presence of reminiscent critical points linked to liquid–liquid phase separations, probably associated with a former upper critical solution temperature point of the respective binary mixtures.

Across the monophasic region of the ternary systems, the curcumin solubility was mapped using UV–vis measurements. In the new ChCl+Lev-based system examined in this study, the solubility of curcumin could be increased by a factor of ~1.5 or 7 compared to the ChCl+Lac and acetone or water systems, respectively. Across the monophasic area of the ternary system, extraction experiments were performed, achieving total curcuminoid yields of ~84%. This is an improvement of 5–16% as compared to the previously studied experiments. Following cycle extraction experiments showed that the extracts could be enriched. A premature saturation occurred due to the loss of the NADES in the system, possibly caused by the adsorption of choline to the cellulose fibers of the plant material. When replacing the consumed choline salt with a fresh one in the solvent mixture, an even higher enrichment of the extracts with curcumin should be possible. In total, this newly studied system has great potential for extracting curcuminoids from the rhizomes of *C. longa*.

During this study, curcumin was investigated as a model compound of a phytochemical that occurs in high amounts in nature. In future research, it would be interesting to examine these ternary systems regarding other naturally occurring phytochemicals, such as piperine, which occurs in white pepper, or other polyphenols such as quercetin or hesperetin. Further, it is worth studying the relation between the structuring in the different domains of the ternary mixtures and the respective solution of phytochemicals. Maybe a “pre-Ouzo” structuring, as it is probable in the system shown in Figure 2a, is always most suited for optimum solubilization.

## Figures and Tables

**Figure 1 molecules-26-07702-f001:**
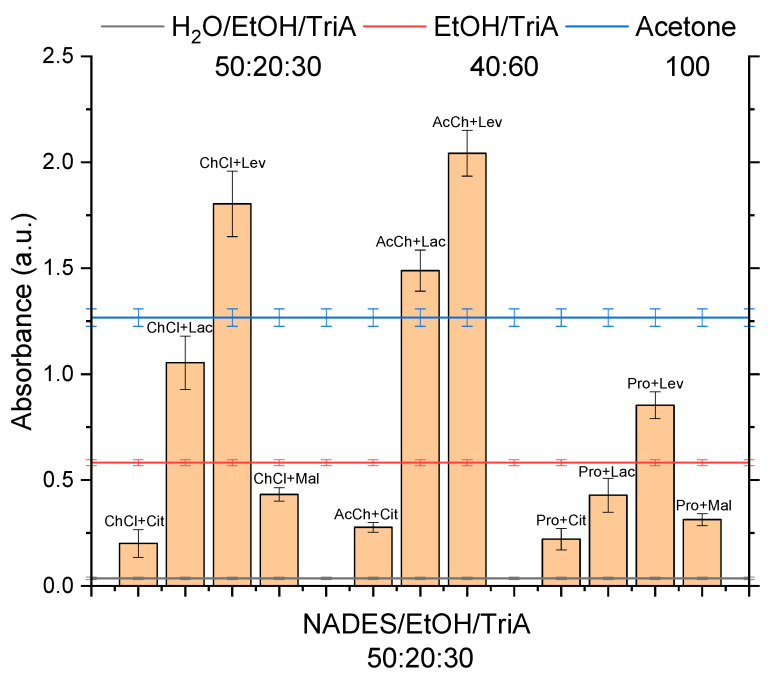
Qualitative solubility of curcumin as determined via UV–vis in a mixture of NADES/EtOH/TriA 50:20:30 in weight. Solubility tests were conducted at room temperature. The reference samples, a ternary mixture of water/EtOH/TriA 50:20:30 (black line at the bottom), the binary mixture of EtOH/TriA 40:60 (red line in the middle, taken from P. Degot et al. [13]), and pure acetone (blue line at the top), are indicated by the horizontal lines.

**Figure 2 molecules-26-07702-f002:**
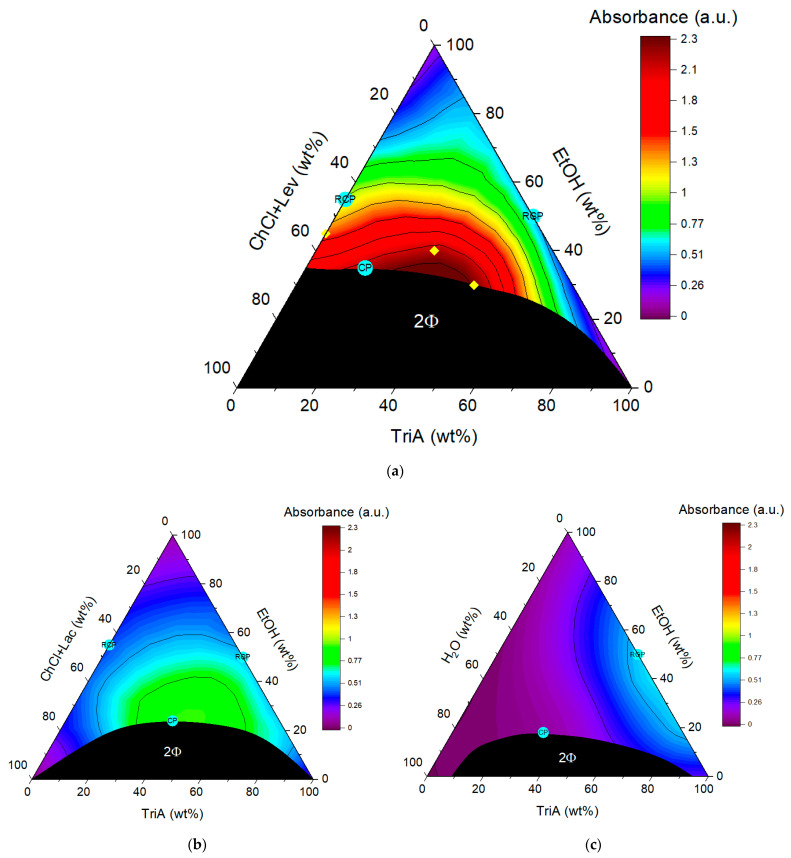
Solubility maps of the ternary phase diagrams consisting of EtOH as the hydrotrope, TriA as the oil phase, and the hydrophilic component (**a**) ChCl+Lev, (**b**) ChCl+Lac, and (**c**) water (with b and c adapted from V. Huber et al. [10]). The black area labeled with 2Φ indicates the region of immiscibility in the ternary systems. The heat maps in the remaining monophasic parts of the diagrams show the solubility of curcumin, where red indicates a high solubility and purple a low one. The yellow diamonds in (**a**) show the compositions where extraction experiments were performed. The turquoise points labeled CP mark the critical points of the ternary systems. The turquoise points labeled RCP represent the reminiscent critical points of the binary mixtures.

**Figure 3 molecules-26-07702-f003:**
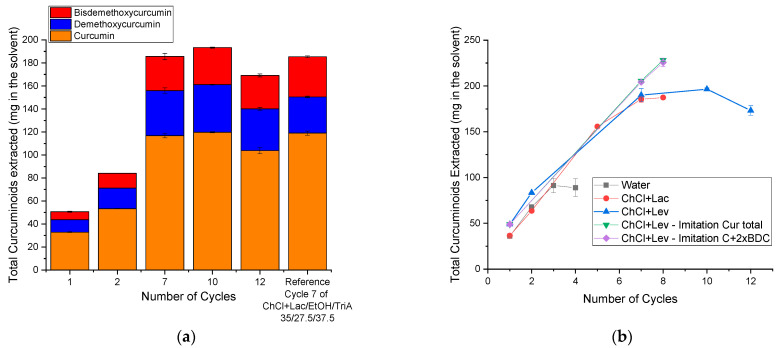
(**a**) Content of all curcuminoids in the extraction solvent after multiple extraction cycles, with the reference being the saturated mixture of ChCl+Lev/EtOH/TriA (35:27.5:37.5) after seven extraction cycles. (**b**) Total curcuminoid content of the extraction systems based on ChCl+Lev (blue triangle), with the references containing ChCl+Lac (red circle [10]), and water (black square [14]). Moreover, the extraction samples prepared with synthetic curcumin (green down-facing triangle) and curcumin + bisdemethoxycurcumin (pink diamond) to imitate the curcuminoid content are presented.

**Figure 4 molecules-26-07702-f004:**
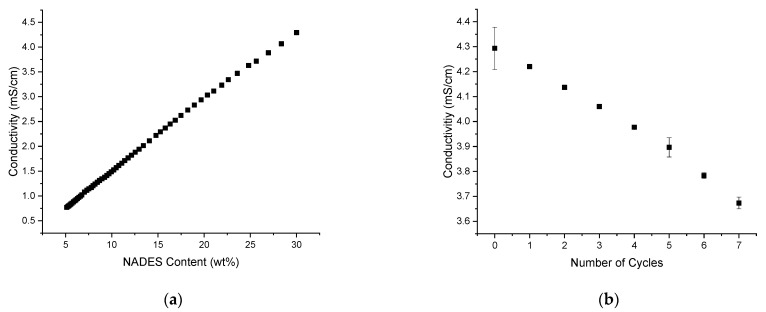
Conductivity measurements of (**a**) the optimum composition of ChCl+Lev/EtOH/TriA (30:40:30) diluted with a binary EtOH/TriA (40:60) mixture and (**b**) of the extraction solutions after multiple cycles of extraction.

**Table 1 molecules-26-07702-t001:** Extraction yields in mg curcuminoid per g rhizome. Results were obtained via HPLC.

Extraction System	Curcumin (mg/g)	Demethoxycurcumin (mg/g)	Bisdemethoxycurcumin (mg/g)
Soxhlet ^1^	11.64 ± 0.70	3.82 ± 0.21	1.67 ± 0.26
Pressure-controlled Soxhlet	14.53 ± 0.62	4.25 ± 0.14	3.82 ± 0.41
H_2_O/EtOH/ TriA 40:24:36 ^1^	9.21 ± 0.32	3.18 ± 0.23	2.89 ± 0.38
ChCl+Lac/EtOH/TriA 35:27.5:37.5 ^2^	11.80 ± 0.36	2.73 ± 0.13	3.26 ± 0.14
ChCl+Lev/EtOH/TriA 25:30:45 Dilution from the optimum binary composition	11.78 ± 0.15	3.94 ± 0.04	2.78 ± 0.05
ChCl+Lev/EtOH/TriA 30:40:30 Optimum Composition	12.04 ± 0.36	4.09 ± 0.13	2.83 ± 0.07
ChCl+Lev/EtOH 55:45	12.13 ± 0.29	4.10 ± 0.11	2.86 ± 0.09

^1^ Data were taken from P. Degot et al. [13]. ^2^ Data were taken from V. Huber et al. [10].

## Data Availability

Not applicable.

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
