# Peer review of "Improvement of the Solubilization and Extraction of Curcumin in an Edible Ternary Solvent Mixture"

_molecules, 2021, doi:10.3390/molecules26247702_

Round 1

Reviewer 1 Report

The manuscript is well organized and presents new and interesting results with high quality level. On my opinion article can be accepted for publication in Molecules as it is.

Reviewer 2 Report

I would like to thank the authors for the nice work entitled “Improvement of the Solubilization and Extraction of Curcumin by Using a Ternary Solvent Mixture of Choline Chloride+Le- vulinic Acid (1:1), Ethanol, and Triacetin.”

Frankly speaking it takes quite a great effort and passion to come up with such good results. More interesting to me was how the authors ruled out the possible reasons for the decrease in the amount of curcuminoids after 7 and 12 cycles of extractions.

I have listed only few concerns as follows:

1. P3L125- “Thus, through these interac-124 tions, a higher solubility of curcumin in NADES mixtures could be reached ….” It's an interesting explanation of the molecular interactions responsible for increasing the yield of the curcumin, however, aren't such interactions a hindrance to complete removal of the NADES (remaining as residual) from the obtained extract?

How do you remove residual ChCl/AcCh if the resulting extract is to be used for analytical/experimental purposes?

2. P4L168- you have stated that “… EtOH is not powerful enough to accomplish a better miscibility of the levulinic acid-based NADES…”

Please explain it further in terms of the structural features and if possible indicate what would be the better solvents to improve miscibility.

Minor

1. P3L107- “In the binary EtOH/TriA mixture a solubility synergy was found in [13] at a compo………”

Please remove in after the word found

2. P13L487- “For the simulation of binary phase diagram consisting of TriA/EtOH, COSMO-RS formulation within the Software package COSMOTherm 2021.”

This sentence seems incomplete please check. 

3. P14L523- “In a concentration range of 0.04 0.2 mg/mL………..”

Please add to between the numbers.

Reviewer 3 Report

The paper entitled "Improvement of the solubilization and extraction of curcumin by using a ternary solvent mixture of choline cholride+levulinic acid (1:1, ethanol and triacetin) is well written and easy to read. The results are new and important for the field of extraction. I only have minor corrections to suggest for the authors before publication:

1) The title is too long.

2)Figure 1: why did you put two times absorbance on teh vertical axis? I think you need to remove one, otherwise it can be confusing for the readers. 

3) Check in the whole manuscript because there are many times H2O instead of H2O.

4) In the conclusion, it would be great to have few sentences about the perspectives of this work. Is it possible to apply the same extraction system to other polyphenols for example?
